# Precision Editing as a Therapeutic Approach for β-Hemoglobinopathies

**DOI:** 10.3390/ijms24119527

**Published:** 2023-05-31

**Authors:** Kiriaki Paschoudi, Evangelia Yannaki, Nikoletta Psatha

**Affiliations:** 1Department of Genetics, Development and Molecular Biology, School of Biology, Aristotle University of Thessaloniki, 54124 Thessaloniki, Greece; paschoudi@bio.auth.gr; 2Gene and Cell Therapy Center, Hematology Clinic, George Papanikolaou Hospital, Exokhi, 57010 Thessaloniki, Greece; eyannaki@uw.edu; 3Department of Hematology, School of Medicine, University of Washington, Seattle, WA 98195, USA

**Keywords:** genome editing, hemoglobinopathies, sickle cell disease, thalassemia, CRISPR/Cas9, base editing, prime editing, hereditary persistence of fetal hemoglobin

## Abstract

Beta-hemoglobinopathies are the most common genetic disorders worldwide, caused by a wide spectrum of mutations in the β-globin locus, and associated with morbidity and early mortality in case of patient non-adherence to supportive treatment. Allogeneic transplantation of hematopoietic stem cells (allo-HSCT) used to be the only curative option, although the indispensable need for an HLA-matched donor markedly restricted its universal application. The evolution of gene therapy approaches made possible the ex vivo delivery of a therapeutic β- or γ- globin gene into patient-derived hematopoietic stem cells followed by the transplantation of corrected cells into myeloablated patients, having led to high rates of transfusion independence (thalassemia) or complete resolution of painful crises (sickle cell disease-SCD). Hereditary persistence of fetal hemoglobin (HPFH), a syndrome characterized by increased γ-globin levels, when co-inherited with β-thalassemia or SCD, converts hemoglobinopathies to a benign condition with mild clinical phenotype. The rapid development of precise genome editing tools (ZFN, TALENs, CRISPR/Cas9) over the last decade has allowed the targeted introduction of mutations, resulting in disease-modifying outcomes. In this context, genome editing tools have successfully been used for the introduction of HPFH-like mutations both in *HBG1/HBG2* promoters or/and in the erythroid enhancer of *BCL11A* to increase HbF expression as an alternative curative approach for β-hemoglobinopathies. The current investigation of new HbF modulators, such as ZBTB7A, KLF-1, SOX6, and ZNF410, further expands the range of possible genome editing targets. Importantly, genome editing approaches have recently reached clinical translation in trials investigating HbF reactivation in both SCD and thalassemic patients. Showing promising outcomes, these approaches are yet to be confirmed in long-term follow-up studies.

## 1. Introduction

Beta-hemoglobinopathies are a group of inherited recessive disorders, caused by a wide range of mutations within the β-globin locus. It is estimated that approximately 400,000 affected births occur annually [1]. Until now, more than 300 mutations affecting the β-globin expression have been described, leading to decreased or absent β-globin production (β^+^ or β^0^ thalassemia, respectively) or the expression of a mutant variant of the β-globin (sickle cell disease).

In β-thalassemia, reduced production of β-globin chains results in an excess of misfolding alpha-globin chains (a matter of quantity) consequently causing erythrocytes’ membrane damage, early apoptosis, and ineffective erythropoiesis. The secondary consequences of these primary defects are anemia, extramedullary hematopoiesis, splenomegaly, and iron accumulation [2]. The conventional therapy of β-thalassemia includes life-long blood transfusions and iron chelation in order to prevent long-term multi-organ damages caused by iron deposition. Sickle cell disease (SCD) is caused by a point mutation in the sixth codon of the β-globin gene, causing a substitution of glutamic acid for valine and subsequent expression of an abnormal form of hemoglobin (a matter of quality), termed HbS. Under hypoxia, HbS is polymerized, generating deformed, rigid, sickle-shaped red cells (sickle cells) entrapped in microvessels. Erythrocyte, and subsequently leukocyte, entrapment in the microcirculation causes vascular obstruction and tissue ischaemia, thus resulting in painful and life-threatening acute vaso-occlusive crises (VOC) [3]. The severity of the disease depends on the co-inheritance of SCD, either with beta-thalassemia mutations that reduce total HbS production, or with the hereditary persistence of fetal hemoglobin (HPFH) that results in high HbF levels with anti-sickling properties in adult life [4]. In the same context, Hydroxyurea (hydroxycarbamide), the only FDA-approved drug for SCD treatment, prevents VOC via HbF induction [5,6].

The only established curative option for both β -thalassemia and SCD is the allogenic transplantation of hematopoietic stem cells (allo-HSCT). The major drawbacks of allo-HSCT are the requirement of an HLA-matched donor, age restrictions for a successful therapeutic outcome (≤14 years), and the necessity for long-term immunosuppression in order to prevent or treat the immunological complications associated with the procedure [7].

In the last 30 years, the development of a wide range of gene therapy approaches for β-hemoglobinopathies offers not only an alternative but also an equally effective, curative option for patients without an available HLA-identical donor [8]. Gene therapy for β-hemoglobinopathies conventionally relies on the autologous transplantation of genetically modified hematopoietic stem and progenitor cells (HSPCs). The procedure includes mobilization of CD34+ HSPCs from the bone marrow to circulation after treatment with pharmaceutical reagents such as the granulocyte-colony stimulating factor G-CSF in combination with Plerixafor (AMD3100; Mozobil™) in thalassemia patients [9,10,11], or Plerixafor alone in SCD patients [12]. Following leukapheresis, the mobilized HSPCs are genetically corrected ex vivo and re-administered to the preconditioned patient [13].

Presently, there are two leading gene therapy approaches. The first and the most well-studied approach is the addition of the therapeutic gene, which could be the β- or γ-globin gene, or a modified β-globin gene (βT87Q, β^AS3^) with anti-sickling properties [14,15,16,17,18]. Gene addition is a “one-size-fits-all” approach; irrespective of the underlying mutation, the normal globin gene that is added into the hematopoietic stem cells (HSCs) expresses the normal protein, and the thalassemic or sickle cell phenotype can be corrected. The insertion of the therapeutic cassette in patients’ genomes to ensure life-long gene expression usually employs an integrating lentiviral (LV) vector. To date, gene addition approaches for thalassemia and SCD have been extensively applied both in preclinical models [17,19,20] and in clinical trials with therapeutic outcomes [21,22,23]. Despite the success, there are still some limitations: (i) the need for high vector copy numbers (VCNs) in order to achieve sustained and increased transgene expression, (ii) the need for optimization of the gene transfer protocols for improved transduction efficiency, (iii) the low engraftment rate of genetically modified cells that is correlated with suboptimal therapeutic outcomes, and (iv) the potential development of insertional mutagenesis phenomena due to the semirandom integration pattern of LV vectors [24].

In contrast to the “one-size-fits-all” gene addition approach, gene editing is a mutation-specific approach. The tremendous development of genome editing technology (e.g., ZFN, TALENs, CRISPR-Cas9) allowed for the targeted introduction of mutations in the genome in order to correct point mutations. Single-mutation diseases such as SCD, represent ideal targets for gene editing. Moreover, the ability of gene editing to introduce targeted mutations that may cause a disease-modulatory effect made it possible to use a universal approach to target, in addition to SCD, the many different thalassemia mutations through the reactivation of the endogenous γ-globin gene [25,26,27,28,29,30,31,32]. Gene editing utilizes for the targeted introduction of point mutations or small deletions and insertions of the cells’ endogenous DNA damage response mechanism. Generally, two main pathways are involved: non-homologous end joining (NHEJ) and homologous derived recombination (HDR) [33,34]. The variety of genome editing tools generates multiple therapeutic avenues, overcoming some concerns as regards gene addition with integrating viral vectors, including insertional genotoxicity and recombination events during viral production [35,36]. In this review, we will discuss and review the genome and precision editing attempts to correct the phenotype of β-thalassemia major and sickle cell disease.

## 2. The Beta-Globin Locus and the Hemoglobin Switching

Hemoglobin, the tetrameric protein responsible for oxygen transportation from the lungs to all tissues, is formed by a symmetric pairing of a globin chain dimer consisting of two alpha-like and two beta-like globin chains. The alpha-globin locus including the α- and ζ-globin genes is located on chromosome 16, while the beta-globin locus is encoded by a gene cluster of ε-, γ-, δ-, and β-globins on chromosome 11. These globin genes are developmentally regulated leading to several forms of hemoglobin during different developmental stages. Specifically, the ζ2ε2, α2ε2, and ζ2γ2 tetramers are present during embryonic development while the α2γ2, also known as fetal hemoglobin or HbF, predominates during fetal life. The switch from fetal to adult hemoglobin, mostly α2β2 (HbA), and in lower frequency α2δ2 (HbA2), occurs shortly after birth during the neonatal period [2,37,38,39]. These developmental switches in the β-globin locus, from embryonic to fetal and fetal to adult hemoglobin, are tightly regulated by the interaction of the beta-globin locus control region (LCR) with the promoters of the respective genes (Figure 1). The beta-globin LCR consists of five DNase I hypersensitive sites (DHSs), of which HS2-4 express a cell type-specific enhancer activity, increasing the transcription levels of each of the β-globin-like genes in a developmental manner [37].

Normally, HbF expression in adult life is reduced to 1%, however, in some cases, fetal hemoglobin expression persists at higher levels. This benign condition is usually caused by mutations in the β-gene cluster or the *γ*-promoter gene region and is known as the hereditary persistence of fetal hemoglobin (HPFH). HPFH is generally classified into two categories: deletional and non-deletional. *Deletional HPFH* is characterized by large deletions (~13–85 kb) in the regions between the β- and γ-globin genes. The majority of these large deletions lead to the loss of δ-globin, effectively altering the dynamics of LCR interaction with γ- and β- genes, and/or the ψβ pseudogene which seems to include a hemoglobin switching-related regulatory element [40,41,42]. Non-deletional HPFH is caused by point mutations or small deletions within both γ-globin *HBG1* (Aγ) and *HBG2* (Gγ) promoters. Some of these SNPs either disrupt transcription suppressors’ binding sites or create de novo binding sites for *HG1/HBG2* transcription activators. For instance, the single nucleotide substitution such as −114 C > T, −117 G > A, 13 bp deletion [Δ13 bp] and −195 C > G, −196 C > T, −197 C > T, −201 C > T, −202 C > T disrupt binding sites of two major HbF silencers, BCL11A and ZBTB7A, respectively [41]. In contrast, specific point substitutions such as −113 A > G, −175 T > C, and −198 T > C create de novo binding sites for the erythroid activators GATA1, TAL1, and KLF1, respectively, that are involved in globin regulation [43,44,45,46] (Figure 1). Co-inheritance of HPFH with β-thalassemia or sickle cell disease ameliorates the severity of the disease and reduces or abrogates blood transfusions or SCD-associated complications [47,48]. The development of genome editing tools has granted the possibility to either attempt correction of the disease mutations or to mimic naturally occurring HPFH or HPFH-like mutations in order to reactivate the silenced γ-globin.

## 3. Genome Editing Tools

In the last two decades, the exuberant development of powerful genome editing tools has permitted the introduction of precise genetic modifications in order to tightly manipulate gene expression (Figure 2). The most widely applied genome editing tools are zinc finger nucleases (ZFN) [49,50,51], transcription activator-like effector nucleases (TALENs) [52,53,54], and clustered regularly interspaced short palindromic repeat (CRISPR)-associated nuclease Cas9 [55,56,57,58]. The common feature among genome editing tools is the accurate introduction of double-strand breaks (DSBs) at the genomic sequence of interest that leads to the activation of endogenous DNA repair mechanisms. Depending on the cell cycle stage, the DSB could be repaired either by non-homologous end joining (NHEJ), the most frequently activated repair mechanism [34], homologous directed repair (HDR), or microhomology-mediated end joining (MMEJ) [59,60]. As briefly described above, NHEJ is an error-prone mechanism that leads to the random introduction of small deletions and/or insertions (indels) to the target genomic region, resulting in frameshift mutations. Comparatively, HDR-mediated mutations are more precise, however, they can only occur in the S and G2 phases of the cell cycle as this pathway requires a DNA sequence as a donor template for DSB healing [33,60].

ZFNs are artificial nucleases composed of two different domains: the DNA recognition domain and the DNA cleavage domain. The DNA recognition domain consists of three or more, individual zinc finger protein subdomains that selectively recognize a three-base pair segment in the target sequence. The enzymatic cleavage of the target DNA sequence is most usually performed by the endonuclease domain of Fok1 [61]. The Fok1 endonuclease domain lacks any DNA recognition properties, therefore the specificity of DNA targeting relies on the accurate interaction between the nucleotide sequence and amino-acid residues of the ZFN DNA-binding domain. Given that double-strand cleavage requires the dimerization of Fok1 endonuclease, a pair of ZFNs attaching to the target sequence is required to introduce the desired editing [49,62].

In a similar manner, TALENs consist of a DNA recognition domain of highly conserved 33–34 amino acid sequences and a Fok1 endonuclease. The specific recognition of the target site is performed by two highly variable positions in each amino acid sequence, position 12 and 13. Amino acids at positions 12 and 13 are commonly referred to as repeat variant di-residues (RVDs) and it has been shown that the four most common RVDs are NN, NG, HD, and NI with a preferential binding affinity towards G/A, T, C, and A, respectively [53,63,64]. Similar to ZFNs, TALENs act in pairs due to the fact that Fok1 performs DSBs only in a dimerized form.

CRISPR-Cas9 is an RNA-guided endonuclease system that was first described as part of the bacterias’ and archeas’ adaptive immunity against foreign DNA [65,66]. The evolution of CRISPR-Cas9 as a novel, multiplexable, and powerful tool for precise genome manipulation, has greatly simplified the application of the targeted genome editing process in a wide spectrum of organisms in order to alter the gene transcription profile [67,68,69,70,71]. The structure of the CRISPR-Cas9 system consists of two components: a single guide RNA (sgRNA or gRNA) and a Cas9 nuclease. The sgRNA is uniquely hybridized with the targeted DNA sequence, usually referred to as “protospacer”, leading to the introduction of a double-strand break by the Cas9 nuclease. The key requirement for this procedure is the recognition of the protospacer adjacent motif (PAM) by Cas9, located near the 3′ end of the protospacer. The PAM sequence recognition varies between different nucleases [56,57]. The Cas9 nuclease derived from *Streptococcus pyogenes* (SpCas9) and the PAM sequence that recognizes, “5′-NGG-3′”, are the most well-studied and widely applicable. The cleavage position of spCas9 is located within the protospacer sequence 3 bp upstream of 5′end of PAM [72]. In contrast to ZFNs and TALENs, the design and use of CRISPR-Cas9 is a relatively simple and less costly procedure [73].

Nuclease-based strategies result in highly efficient genome editing, both in ex vivo and in vivo approaches; however, the introduction of DSBs has been correlated with severe adverse effects. CRISPR-Cas9-induced DSBs lead to DNA damage responses and activation of the p53 damage response pathway [74,75,76]. Furthermore, the simultaneous introduction of DSBs affects chromosomes’ integrity resulting in chromosomal translocations and chromothripsis, a catastrophic event of massive rearrangements occurring “all-at-once” and leading to a cascade of genome instability and potential oncogenesis [77,78].

Recently, the development of the *base editing (BE)* technology seems to be a safer and equally effective genome editing approach. Base editing allows the targeted incorporation of a specific point mutation, bypassing DSBs and HDR-mediated genome editing. In base editors (BEs), a catalytically deactivated form of Cas9 (dCas9) or a nickase-Cas9 (Cas9n) is fused with a deaminase that selectively deaminases particular DNA bases. There are two categories of base editors according to the type of enzyme that catalyzes the deamination: cytidine base editors (CBEs) and adenine base editors (ABEs) [79,80]. CBEs rely on a cytidine deaminase that converts the desired cytosine to uracil after the hybridization of the spacer sequence with the gRNA. The endogenous DNA repair mechanism recognizes uracil as a thymidine resulting in C-G to T-A transition [80,81,82]. ABEs catalyze the inversion of the A-T pair to the G-C pair in the DNA duplex. A natural enzyme catalyzing adenosine deamination has yet to be discovered; the only adenosine deamination phenomena observed include cases of free adenine, free adenosine, adenosine in RNA, or adenosine in mispaired RNA–DNA heteroduplexes. Gaudelli et al. created an engineered ABE after the fusion of a dCas9 with a tRNA adenosine deaminase enzyme, TadA, originated from Escherichia coli [83]. The TadA enzyme catalyzes the hydrolytic deamination of adenosine, converting it to inosine that is recognized by the cellular repair mechanism as guanosine, and the intermediate T-I base pair is replaced by a G-C base pair [83,84,85,86]. Compared to cytosine base editing, editing with ABEs is much “cleaner”, without significant percentages of indels or off-target effects [87].

Another promising approach to overcome previous CRISPR-Cas9 limitations are the recently developed *prime editors (PEs).* While they share some common features with regular CRISPR-Cas9, PEs have additional properties that increase the specificity and safety of the approach. Avoiding double-strand brakes, and in contrast to HDR-mediated genome editing employed by BEs, the utilization of PEs efficiently allows the incorporation of all of the possible combinations of base substitutions [80]. The major difference between PEs and other Cas9 modules is that the gRNA used is not only complementary to the target but also contains an RNA template sequence, the prime editing guide RNA or pegRNA. PEs consist of a Cas9n (H840A) that nicks the non-complementary strand three bases upstream of PAM. Additionally, the Cas9n is fused with an RNA-dependent DNA polymerase derived from Moloney murine leukemia virus reverse transcriptase (M-MLV RT) that is responsible for the reverse transcription of the RNA template and eventually for the introduction of the desired alteration in genome [79,80,88,89,90].

## 4. In Situ Mutation Correction

### Sickle Cell Disease

Genome editing platforms, including ZFNs, TALENs, and CRISPR-Cas9, can be used for HDR-mediated genome editing to correct the disease mutation, using a DNA-donor sequence as a template. For SCD specifically, in which a single point mutation is responsible for the disease phenotype, the development of an HDR-based genome editing approach in order to replace the mutant *HBB* gene can be widely applied to all patients. HDR is a high-fidelity mechanism, however, as it is mainly active in the S/G2 cell cycle phase, the HDR-mediated gene correction in the quiescent HSCs has an overall poor efficiency [91].

Codelivery of ZFNs with a DNA donor template in SCD patient-derived inducible pluripotent stem cells (iPSCs) has been shown to correct the SCD mutation but with low-efficiency [92,93]. Hoban et al. used ZFNs in combination with a single-strand oligonucleotide (ssODN) donor template or an integrase-defective lentiviral vector (IDLV) to replace the mutated region with a wild-type sequence in patient-derived HSPCs. HDR-mediated correction increased HbA levels in corrected HSPCs-derived erythrocytes without affecting the differentiation potential of the HSPCs [26]. A CRISPR-Cas9 gene-editing system that combined Cas9 ribonucleoproteins (RNPs) with an adeno-associated viral vector (rAAV6) for the delivery of a homologous *HBB* donor, resulted in >90% targeted integration and efficiently increased expression of the wild-type β-globin gene in patient HSPCs-derived erythrocytes [94]. In order to increase HDR efficacy, several HDR enhancement approaches have been explored, including tight regulation of DNA repair pathway components or improvements of the genome editing machinery, or alteration of the intracellular environment around the DSBs [95]. More recently, Li et. developed an in vivo genome editing approach, based on the in vivo targeted delivery of a prime editor to HSPCs by the helper dependent adenoviral vector 5/35++(HDAd5/35++). Correction of the SCD mutation using prime editing resulted in enhanced production of HbA and remarkable amelioration of disease phenotype in an SCD mouse model [96] (Appendix A).

## 5. Beta Thalassemia

Contrary to the seemingly straightforward correction approach for SCD, mutation correction in β-thalassemia is highly complex and less widely applicable given the several hundred of different β-thalassemia mutations described to date. Nevertheless, genome editing strategies have also been applied for the correction of the more frequent β-thalassemia point mutations. The CD41/42ΔTTTC is the most common β-thalassemia mutation in Southeast Asia and Southern Asia. HDR-mediated editing using CRISPR-Cas9 and a ssODN harboring the wild-type sequence, efficiently corrected that mutation in β-thalassemia patient-derived iPSCs [97]. The *HBB* IVS2-654 (C  >  T) mutation is the most common mutation in East Asia, known to create an aberrant 5’ splice site within intron2 of *HBB* premature mRNA [98,99]. Both CRISPR-Cas9 and TALENs have been efficiently applied for targeted correction of this mutation via homologous recombination, in β-thalassemia patient-derived iPSCs [100] and a β-thalassemia mouse model [101]. Xu et. al. efficiently disrupted an aberrant splice site using Cas12a/Cpf1 RNPs in patient-derived HSPCs resulting in increased wild-type β-globin expression and HbA formation. For the *HBB* IVS1-110 G > A mutation, which leads to abnormal splicing [102], TALEN or CRISPR-Cas9 mediated editing provided on-target indel frequency up to 95% with subsequent restoration of normal splicing and elevation of the wild-type β-globin expression [103,104]. Interestingly, the utilization of the CRISPR-Cas9 system for the correction of β039-globin mutations in combination with the rapamycin-induced gamma globin reactivation significantly increased the de novo expression of both HbA and HbF [105] (Appendix A).

## 6. HbF Reactivation by Recapitulating Natural HPFH Mutations

Targeted introduction of point mutations or large deletions in the *HB1/HBG2* locus in order to mimic naturally occurring HPFH mutations represents a less personalized procedure compared to a specific mutation correction. Specifically, TALEN-mediated disruption of regulatory elements within *HBG1* and *HBG2* promoters in human CD34+ cells efficiently generating indels (*HBG1* 43% and *HBG2* 74%) resulted in elevated HbF levels, both in vitro and in vivo, after transplantation of edited cells in a humanized mouse model [25]. Furthermore, the CRISPR-Cas9 system has been extensively used for the introduction of HPFH-like mutations. CRISPR-Cas9-mediated 13 kb deletion mimicking the naturally occurring Sicilian HPFH mutation led to a two-fold γ-globin mRNA increase in HSPCs-derived clonogenic erythroid colonies [106]. Disruption of HbF regulatory binding sites proximal to *HBG1* and *HBG2* promoters or deletion of a 13.6 kb genomic region involving β- and δ-globin genes, resulted in sufficient HbF expression in SCD patient-derived HSCs and prevention of sickling [107,108,109]. In vivo, CRISPR-Cas9 mediated the disruption of the BCL11A binding site upstream of gamma-globin genes in β-YAC/CD46tg mice, efficiently increased HbF levels, and improved thalassemic phenotype [110]. Utilization of the cytidine base editor (hA3A-BE3) to introduce HPFH-related mutations at −115 C or −114 C within the *HBG1/2* promoter significantly increased HbF levels in healthy and thalassemic patient-derived HSPCs after ex vivo erythroid differentiation [111]. In a recent study, in vivo, the introduction of −133 A > G substitution in β-YAC/CD46tg mice after safely delivering an ABE-HDAd5/35++ vector, resulted in elevated γ-globin levels [112]. Correction of IVS1-110 (G > A) mutation by the adenine base editor, SpRY-ABE8e caused a ~80% gene correction and increased β-globin production, both ex vivo and in vivo in a beta-thalassemia humanized mouse model [113] (Appendix A).

## 7. HbF Reactivation by Modulating γ-Globin Modifiers

### 7.1. BCL11A and Its Erythroid Enhancer

The B cell leukemia/lymphoma 11A (BCL11A) protein is a C2H2 transcription factor expressed both in hematopoietic tissues (bone marrow, splenic B- T-cells, monocytes, and megakaryocytes) and the brain [114,115]. The *BCL11A* gene is located on chromosome 2 and malfunctioning of *BCL11A* has been correlated with the development of B-cell chronic lymphocytic leukemia and other types of B-cell malignancies [116,117]. It is known that BCL11A regulates B-lymphopoiesis via a p53-dependent pathway that tightly controls apoptosis and cell survival [118]. In vivo experiments revealed a detrimental role of *BCL11A* gene ablation in B-lymphopoiesis; the knockout of *BCL11A* in mice resulted in impaired B-lymphopoiesis and perinatal death [119]. Additionally, reduced expression of BCL11A in HSCs has been correlated with the presence of an aging-like phenotype [120].

Genome-wide association studies (GWASs) also implicated BCL11A as a major modulator for the HbF expression. Particular single-nucleotide polymorphisms (SNPs) in the BCL11A locus have been associated with elevated levels of γ-globin expression and subsequent HbF formation [121,122,123], ameliorating the β-hemoglobinopathies’ phenotype [124,125]. Extensive in vitro and in vivo studies elucidated the key role of BCL11A in the developmental regulation of the γ-globin expression [126,127]. Furthermore, partial inactivation of BCL11A via RNA interference (RNAi) or erythroid conditional knockout of *BCL11A* efficiently reactivated fetal globin expression in human and murine erythroid cells [127,128,129]. Even though BCL11A is expressed in several subpopulations of blood cells, it has been observed that in erythroid cells its expression is tightly regulated by an erythroid-specific enhancer, located in the second intron of the *BCL11A* gene. Specifically, three DNase hypersensitive sites (DHSs) in 2-intron (+55, +58, and +66) were found to exhibit gene regulation properties [130] while SNPs within this erythroid enhancer seem to prevent the binding of GATA1 and TAL1 [131,132], principal transcription factors of erythroid commitment and differentiation [133]. High-resolution analysis of the β-globin locus via ChIP-seq and CUT&RUN indicated that BCL11A preferentially binds TGACCA motifs within the γ-globin promoters disturbing interactions with the β-LCR and blocks γ-globin expression. Furthermore, mutations in TGACCA motifs have been also observed in individuals with HPFH [42] (Appendix A).

BCL11A has become a preferred target for precise genome editing in the field of β-hemoglobinopathies gene therapy. Given that the absence of BCL11A expression could have detrimental effects on cells’ viability and organism development [118,120] the discovery of the erythroid-specific enhancer of *BCL11A* offered new opportunities for tissue-specific BCL11A inactivation. In this context, a variety of studies have been performed to evaluate the safety and efficiency of genome editing-mediated disruption of *BCL11A* enhancers. In one of the first reports using patient cells, ZNF-mediated disruption of the *BCL11A* erythroid enhancer in HPSCs derived from thalassemic patients resulted in elevated levels of γ-globin expression at the end of an erythroid differentiation culture, improvement of erythroid cell phenotype and maturation and selective survival of corrected cells [27]. In addition, TALEN-mediated disruption of 2-intron of *BCL11A* in CD34+ cells of non-human primates (NHP) efficiently increased γ-globin expression in ex vivo differentiated NHP CD34^+^cells while the autologous transplantation of *BCL11A*-edited CD34^+^cells in NHPs resulted in low levels of editing rates that subsequently led to mild, yet prolonged HbF expression [134]. CRISPR-Cas9-mediated disruption of the GATA1 binding site on DHS +58 selectively decreased BCL11A expression in erythrocytes leading to increased HbF levels and improved phenotype of SCD HSPCs-derived erythrocytes. Importantly, genome editing did not influence the modified cells’ engraftment ability and differentiation potential after transplantation into NOD.Cg-Kit^W−41J^ Tyr ^+^ Prkdc^scid^ Il2rg^tm1Wjl^ (NBSGW) mice and is currently being tested in a clinical trial [135,136]. To further evaluate the potential toxicities derived from genome editing in a large animal model, Demirci et al. transplanted *BCL11A* enhancer-edited rhesus HSPCs and reported no effect on engraftment rate. Importantly, follow-up after 100 weeks revealed that potentially therapeutic levels of HbF were still observed in PB RBCs without toxicity events related to gene editing [137] (Appendix A).

Another interesting approach includes the simultaneous disruption of the BCL11A binding site in the *HBG1/HBG2* locus and the erythroid-specific *BCL11A* enhancer with a multiplexed CRISPR-Cas9. HbF reactivation by targeting both sites was superior to either target alone without adding to cell toxicity or off-target events [138]. By incorporating this bimodular CRISPR-Cas9 system into a novel helper dependent adenoviral vector (HDAd5/35++), which uniquely binds to the hCD46 receptor expressed on the surface of human HSCs, the in vivo multiplexed genome editing of human thalassemic HSCs while they were circulating in PB of a humanized NBSGW mouse model of thalassemic hematopoiesis post mobilization, resulted in increased levels of HbF in GlyA+ cells of chimeric bone marrow and subsequent improvement of the thalassemic phenotype [138]. In another study, utilization of the recently developed BE system (A3A(N57Q)-BE3) to introduce a point mutation on the +58-DHS of the *BCL11A* erythroid enhancer reestablished the globin chain balance in both thalassemic and SCD patient-derived HSPCs and reduced sickling of SCD-derived HSPCs [139]. Notably, the introduction of clustering mutations 200 bp upstream of *HBG1/2* genes by CBE or ABEs resulted in HbF reactivation in SCD HSPCs by either the disruption of LRF binding sites or the recruitment of KLF1 transcriptional activator [140]. Furthermore, simultaneous targeting of the *BCL11A* erythroid enhancer and introduction of the common *HBB*-28A > G promoter mutation, by A3A(N57Q)-BE3 efficiently improved the β-thalassemia phenotype [139] (Appendix A).

### 7.2. Other HbF Modifiers

ZBTB7A, also referred to as leukemia/lymphoma-related factor (LRF) or Pokemon, is a zinc finger transcription regulator factor that is broadly expressed in hematopoietic cells and a key molecular regulator in B, T, and erythroid cells [141]. It is known that during erythropoiesis ZBTB7A is activated by GATA1 and subsequently suppresses the pro-apoptotic factor BCL2-like 11 (BIM) preventing erythroid cells’ apoptosis [142]. Interestingly, ZBTB7A acts as a suppressor of HbF expression independently of BCL11A, via recruitment of the NuRD repression complex upstream of *HBG1/2* genes after its binding to −200 position [143]. Mutations within the ZBTB7A binding site have been associated with the development of HPFH. Reduced expression of ZBTB7A resulted in elevated levels of HbF in HUDEP cells and in CD34 + HSPC-derived erythroblasts [41]. Furthermore, CRISPR-Cas9 disruption of the ZBTB7A binding sequence within the *HBG1/HBG2* promoters in SCD patient-derived HSPCs efficiently reactivated γ-globin expression, restored globin chains imbalance and improved SCD phenotype [144] (Appendix A).

Krüppel-like factor 1 (KLF-1) is a well-studied erythroid transcription factor and a chief component of human erythropoiesis [119] known for its involvement in β-globin synthesis and transcription of other erythroid genes (i.e., heme biosynthesis enzyme, blood group antigens) [145,146,147]. Interestingly, specific mutations in the coding region of KLF-1 have been associated with high levels of HbF [148]. CRISPR-Cas9-genome editing has been extensively applied for *KLF-1* disruption in erythroid cell lines, normal- and SCD patient-derived CD34^+^cells which resulted in reactivation of γ-globin expression [149,150,151]. Recently, generation of the specific −198 T > C HPFH mutation was shown to create a binding site for the KLF-1 transcriptional activator, resulting in approximately 60% base conversion and 3.5-fold increase in γ-globin expression in normal donor HSPCs-derived erythrocytes [152].

SOX6 is a chromatin-association protein that belongs to the high mobility group (HMG) transcription factors that bind to the minor groove of DNA altering local chromatin architecture and induces interactions with transcription factors [153,154]. In mice, SOX6 has been characterized as a regulating factor of defined erythropoiesis and silencer of mouse embryonic globins [155,156]. In humans, during erythroid maturation, SOX6 cooperates with BCL11A and GATA-1 in the γ-globin silencing [157]. ZFN or CRISPR-Cas9 mediated disruption of *SOX6* in the erythroleukemic K562 cell line drove the activation of the γ-globin expression [158,159].

In recent years, novel modulators of HbF expression have emerged. This category includes ZNF410, CHD4, and other NuRD components. ZNF410 is a zinc finger transcription factor that uniquely induces *CHD4* (a NuRD nucleosome remodeler) expression which subsequently results in γ-globin transcription suppression [160,161]. In contrast to BCL11A and other well-known HbF transcriptional silencers, ZNF410 induces γ-globin silencing indirectly. Disruption of either *ZNF410* or *CHD4* coding sequences or ZNF410 binding sites upstream of *CHD4* via CRISPR-Cas9, increased HbF levels in HUDEP cells [160,162]. Heme-regulated eIF2α kinase, also termed heme regulatory inhibitor (HRI), is known as an erythroid-specific heme-sensory protein, that is activated under low heme concentration [163,164] and plays a vital role in γ-globin silencing via indirect activation of *BCL11A*. Specifically, after its activation, HRI phosphorylates the eIF2a translation factor and induces the ATF4 transcription factor that uniquely binds to the +55 position of *BCL11A* erythroid enhancer stimulating *BCL11A* transcription. Interestingly, the HRI-eIF2a-ATF4-BCL11A axis uniquely appeared in humans and not in mice [165,166]. CRIPSR-Cas9 mediated disruption of *HRI* or *ATF4* in HUDEP-2 cells increased HbF expression [165]. HIC2 is a zinc finger DNA binding protein that has been recently discovered as a novel HbF regulator. Overexpression of *HIC2* inhibited *BCL11A* transcription via direct binding to the +55 DHS of the erythroid enhancer, reducing chromatin accessibility and H3K27 acetylation. Overexpression of *HIC2* in SCD patient derived-HSPCs increased HBG1/2 mRNA and prevented sickling [167,168]. Following CRISPR-Cas9-based genetic screening of all serine/threonine protein phosphatases, PPP6C, the catalytic subunit of protein phosphatase 6, was detected as a novel HbF suppression. Targeted inactivation of *PPP6C*, via CRISPR-Cas9, in HSPCs derived from normal donors and SCD patients resulted in increased levels of HbF, measured by HPLC, and reversal of SCD phenotype [169] (Appendix A).

## 8. Current Clinical Trials

Genome editing approaches for β-hemoglobinopathies recently have been applied in a clinical trial setting. ZFN was the first genome editing tool used in a clinical trial [NCT03432364; ST-400-01, Sangamo Therapeutics, and Sanofi] for the *BCL11A* erythroid enhancer disruption in thalassemic and SCD- patient-derived CD34^+^cells. Following a six-month follow-up, rapid hematologic reconstitution and persistence of editing rates that led to increased HbF expression were observed in SCD patients who remained symptom-free at 52 and 32 weeks and 29 days post-transplantation of CD34^+^-edited cells [170]. In parallel, two-phase 1/2 clinical trials, funded by VERTEX-Therapeutics in collaboration with CRISPR-Therapeutics, based on CRISPR-Cas9 disruption of *BCL11A* erythroid enhancer in CD34^+^cells derived from transfusion-dependent thalassemic (TDT) patients [NCT03655678; CTX001-111] and SCD-patients [NCT03745287; CTX001-121], demonstrated that 42 of 44, TDT-patients were free of transfusion for up to 36 months, maintaining after month 3 total Hb levels > 11 g/dL of which more than 90% was HbF while the 31 SCD patients remained severe vaso-occlusive crises (VOCs)-free for a maximum duration of 32 months, maintaining stable levels of approximately 40% HbF [171,172]. In another clinical trial, the disruption of the GATA1-binding site at the +58 position of *BCL11A* erythroid enhancer, in two pediatric patients with β0/β0 and β+/β+ thalassemia, resulted in transfusion independence in both, with persistent editing levels equal or above 85% which correlated with Hb levels of 14–15 g/dl (NCT04211480) [136]. The single-cell transcriptomic analysis did not demonstrate substantial transcriptional changes by the therapeutic intervention in non-erythroid cells. Τwo additional clinical trials enrolling SCD patients, performed by Graphite Bio, Inc [NCT04819841; GPH101-001; CEDAR] and Editas Medicine, Inc [NCT04853576; EM-SCD-301-001], target ex vivo BCL11A disruption in patients derived CD34+ cells via CRISPR-Cas9 system [173] (Appendix A). 

## 9. Remaining Challenges

After decades of research, the recent advances in technologies manipulating the human genome fueled by the improved understanding of globin gene regulation, have drastically changed the therapeutic landscape for hemoglobinopathies by including various genetic treatments with high curative potential. Hematopoietic stem cell gene therapy by gene addition and genome editing have both provided remarkable clinical results in transfusion-dependent β-thalassemia and SCD. Studies are underway to define the long-term safety and sustainability of the clinical benefit provided by the two approaches.

Gene editing, through precision targeting, has been considered a safer approach than the gene addition strategy by overcoming the semi-random integration issue of retroviral vectors and the potential of insertional mutagenesis. However, off-target genotoxicity has become evident with gene editing due to unintentional cleavage at genomic loci with high homology to the target sequence, potentially leading to additional point mutations, deletions, insertions, or even chromosomal rearrangements and translocations [174,175,176]. The development of high-fidelity versions of Cas9 for increased specificity for the *HBB* locus has significantly reduced DNA modifications at unintended genomic sites [177] while several tools and assays have been developed to predict and detect off-target effects [176,178,179,180]. While the off-target toxicity has now been well-studied, detected, and predicted, the recently described and previously unappreciated on-target genotoxicity resulting in large deletions, truncations, inversions, aneuploidy, and even chromothripsis [77,181,182], cannot be reasonably reduced by further increasing the specificity of DSB approaches. The diversity and complexity of these large genomic alterations up to a megabase-scale magnitude, requires a combinatorial detection strategy as existing single-assay approaches cannot detect them all. As many of the initial on-target events can be tumorigenic and become evident after being amplified into extensive genetic alterations in subsequent cell cycles, long-term safety studies post the completion of patient follow-up in the parental clinical trials are indispensable.

Another previously unappreciated hurdle for the transitioning of gene therapy for hemoglobinopathies from clinical trials to clinical practice has been the exorbitant costs of commercialized gene therapy products. The 1.5 million € price of the first gene addition therapy product approved for β-thalassemia (betibeglogene autotemcel or beti-cel, Zynteglo, BlueBirdBio) led to reimbursement negotiation failure and its commercialization arrest in Europe. Strikingly, beti-cel has been granted a recent FDA approval [183], at a price tag of $2.8 million, becoming the most expensive medicine in history. Such prices are prohibitive as upfront costs for public and private payers even in robust national economies while undoubtedly unaffordable in low-to-middle-income countries, thus precluding the access to treatment of those in need and deepening inequality between patients. Gene editing for hemoglobinopathies is predicted to soon be at the stage of commercialization and as also with this platform, the manufacturing, the supply chain, and patient care remain complex, and a very high price cost is again anticipated. A global framework needs to be urgently established on pricing issues of HSC gene therapy products, based on global and open discussions among all the stakeholders involved, on the actual development and manufacturing costs against the savings anticipated by a one-time treatment, sparing life-long supportive care and numerous, previously lost, productivity hours.

In vivo HSC gene therapy/editing by transducing mobilized CD34^+^ cells directly in peripheral blood by injecting HSC-tropic viral vectors when the circulating HSPCs are in their highest concentration, represents a novel, simplified, and minimally invasive platform that avoids leukapheresis, ex vivo cell manipulation, myeloablation, and transplantation [184]. This “portable” approach, developed by the lab of A. Lieber, has been shown to safely and efficiently transduce primitive HSCs and ameliorate or correct thalassemia or SCD phenotypes in human HSCs, mouse models, and non-human primates, using either gene addition systems or gene editing tools [110,185,186]. It is expected that, upon definitive optimization, it may represent an affordable and accessible gene therapy treatment even in resource-poor regions where hemoglobinopathies are endemic and HSPC transplantation is not feasible.

## Figures and Tables

**Figure 1 ijms-24-09527-f001:**
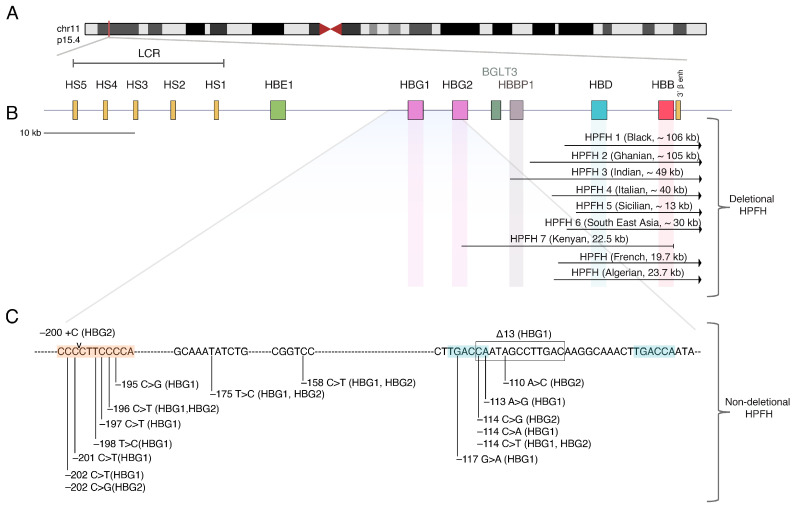
Hereditary persistence of fetal hemoglobin (HPFH). (**A**) The beta-globin locus is located at p15.4 of chromosome 11 and consists of (**B**) the β- locus control region (LCR), hypersensitive sites (HS1-HS5), and beta-like genes family, including *HBE1*, *HBG1*, *HBG2*, *BGLT3*, *HBBP1*, *HBD*, and *HBB.* The most common HPFH-related deletional mutations caused by deletions of large genomic sequence fragments are depicted. (**C**) Non-deletional HPFH mutations (nucleotide substitutions or small deletion) are located within the promoters of *HBG1* and/or *HBG2*. Highlighted are the sequences recognized and bound by the transcriptional HbF regulators; LRF (red), and BCL11A (blue).

**Figure 2 ijms-24-09527-f002:**
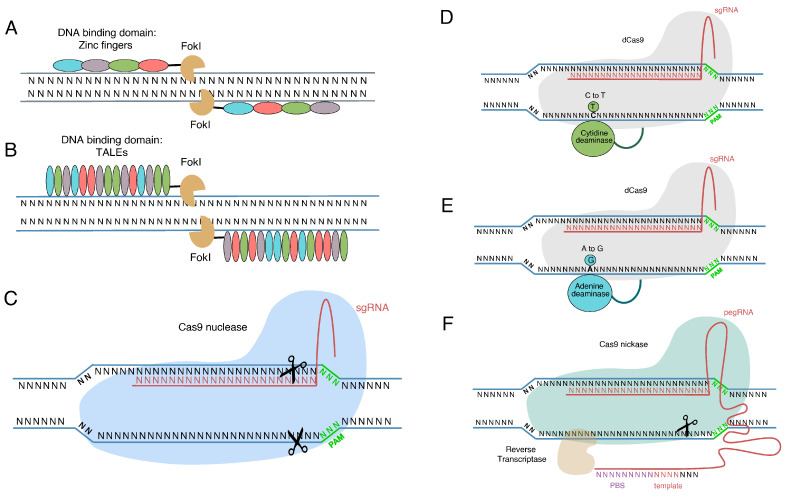
Genome editing platforms. (**A**) Zinc finger nucleases (ZFNs); (**B**) transcription activator-like nucleases (TALENs); (**C**) clustered regularly interspaced short palindromic repeats (CRISPR)/CRISPR associated protein 9 (Cas9); (**D**) cytidine base editor (CBE); (**E**) adenine base editor (ABE). (**F**) Prime editor (PE) sgRNA: single guide RNA, pegRNA: prime editor guide RNA, PAM: Palindromic Adjacent Motif, dCas9: dead-Cas9 nuclease, PBS: primer binding sequence.

## Data Availability

Not applicable.

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
