# Peer review of "Precision Editing as a Therapeutic Approach for β-Hemoglobinopathies"

_ijms, 2023, doi:10.3390/ijms24119527_

Round 1
Reviewer 1 Report
The manuscript “Precision editing as a therapeutic approach for beta-hemoglobinopathies” by Paschoudi and co-authors present a clear and comprehensive review on the approaches for genome editing for the sake of hemaglobinopathies treatment. The review contains a description of the beta-globin locus, its genes and control regions, as well as a catalogue of mutations found in this locus. In the following section, a brief description of gene editing tools is made. Finally, authors present a compilation of current approaches to use gene editing to cure hemaglobinopathies, such as sickle cell disease and β-thalassemia. The manuscript contains a discussion of the tools which are at a stage of clinical trials or approval by the regulators. In general, the manuscript is a well written focused review.
I suggest a few corrections related to gene editing tool section.
Line 231: “…resulting in C-G to A-T transition”. To my knowledge, C is converted to U which is equivalent to T, so the fragment should be reformulated as follows: “…resulting in C-G to T-A transition”.
Line 239: “… inosine that is recognized…as a cytosine and the intermediate T-I base pair is replaced by a G-C base pair.” Inosine has basepairing properties of guanosine, thus the sentence should read: “… inosine that is recognized…as a guanosine and the intermediate T-I base pair is replaced by a C-G base pair.”
Line 245: “Avoiding the double strand brakes, in contrast to BEs”. Base editing goes without double strand breaks. It is HDR-dependent repair following Cas9 cleavage that goes via double strand breaks. The text should be rephrased accordingly.
Reviewer 2 Report
This review is mainly focused on the gene editing technology as the therapeutic approach for the inherited recessive hemoglobinopathies disease. The manuscript is well-written and organized. Author describe about the beta hemoglobinopathies, the root cause of the disease, conventional treatment approaches, FDA approved drug etc in details.
Further author explained all the gene therapy approaches, gene editing tools and their mode of actions. Author also described current clinical trials, challenges and about associated therapeutic companies and investment.
In my review, the manuscript is well organized, comprehensively described and the content is scientifically sound. The manuscript is suitable for publication in IJMS however after minor modification to this current version.
Below are few minor comments author should address in their revised manuscript.
1. Author could improve the quality of the figures.
2. Page 7, lines 277-285 and page 10, lines 449-453
Text font is different from rest of the manuscript.
3. Author can also include a few recent references in the field
Biology 2022, 11, 862. https://doi.org/10.3390/biology11060862
J. Clin. Med. 2021, 10, 482. https://doi.org/10.3390/jcm10030482
Genes 2022, 13, 1727. https://doi.org/10.3390/genes13101727
https://doi.org/10.1016/j.ymthe.2021.10.001
Reviewer 3 Report
The manuscript by Paschoudi K et al reviews various gene editing strategies targeting beta-hemoglobinopathies.
Overall, the manuscript provides a good coverage of the field with a balanced description of strength and limitations of specific tools as applied to various hemoglobinopathies.
However, there is a glaring lack of a table summarizing the literature on all the gene editing strategies applied to beta-hemoglobinopathies in vitro as well as ex vivo. The readers will greatly benefit by inclusion of such table in this manuscript.
Another table summarizing the results of clinical trials on gene editing tools with respect to beta-hemoglobinopathies, including the ones which are approved in various countries would greatly improve the quality of the manuscript.
Sentences are sometimes too long to be clear. At many places, the use of grammar is not appropriate.
Round 2
Reviewer 3 Report
The authors have adequately addressed all the concerns in the revised version of the manuscript.
The quality of English has been significantly improved in the revised version of the manuscript. Minor checks might be needed.